# Relationships between Climate Variability and Radial Growth of *Larix potaninii* at the Upper Altitudinal Limit in Central Hengduan Mountain, Southwestern China

Haitao Yue [1,2], Jianing Li [3], Siyu Xie [1], Hai Chen [4], Kun Tian [1], Mei Sun [1], Dacai Zhang [2] and Yun Zhang [1,*]

1    National Plateau Wetlands Research Center, Wetlands College, Southwest Forestry University, Kunming 650224, China; yuehaitom@163.com (H.Y.); xiesy_d@163.com (S.X.); tlkunp@126.com (K.T.); sm0510215@163.com (M.S.)

2    College of Biodiversity Conservation, Southwest Forestry University, Kunming 650224, China; dczhang24@163.com

3    Institute of Forestry Engineering, Guangxi Ecoengineering Vocational & Technical College, Liuzhou 545004, China; lijianing0411@163.com

4    Changxing Forestry Technology Extension Center, Huzhou 313100, China; boy840416@163.com

*    Correspondence: zhangyuncool@163.com; Tel.: +86-136-2967-0659

**Abstract:** Improved understanding of the responses of stem radial growth to climates is necessary for modeling and predicting the response of forest ecosystems to future climate change. We used dendrochronological methods to study climate effects on the radial growth of a subalpine deciduous conifer, *Larix potaninii*. Tree-ring residual chronologies were developed for five sites at the upper distributional limits in the Central Hengduan Mountains, Southwestern China. Redundancy analysis and response function were used to compare inter-annual variability in growth sensitivity among the chronologies and to identity key climatic factors controlling tree radial growth. The results showed that both precipitation and temperature influenced tree growth, and response patterns were consistent for five chronologies. During the current year's early growing season ($T_{mean}$ in May and $T_{max}$ in June), temperature positively affected the radial growth of *L. potaninii*, while September $T_{min}$ and October precipitation in the previous year and May and June precipitation in the current year all had negative impacts on its radial growth. *L. potaninii* growth appeared to be mainly limited by photothermal conditions in May and June. In the context of increasing $CO_2$ concentrations accompanied with warmer temperatures, future climate change would likely stimulate the radial growth of *L. potaninii* in Central Hengduan Mountain.

**Keywords:** climate warming; dendrochronology; climatic response; subalpine forest; Northwestern Yunnan Plateau; deciduous conifer

## 1. Introduction

The species distributional limit is believed to be physiologically controlled by low temperatures, implying that an extreme distribution area should be highly sensitive to global warming [1]. For the area of the Central Hengduan Mountains (CHMs), recent research indicated a trend of significant warming during the past few decades at a rate of 0.3 °C/decade [2]. Furthermore, tree-ring climate reconstructions in the CHMs showed a warmer summer [3] and increases in the annual mean temperature since the 1990s [4], as well as a declining trend in spring humidity between 1945 and 2011 [5]. Based on conventional explanations of tree growth at the treeline, one would predict increased radial growth at the upper distributional limit in the CHMs under the background of continued warming [6].

Tree radial growth for a given year illustrates the impact of environmental conditions, among which, climatic variables are important factors affecting tree growth [7]. Tree-ring data often contain a complex set of potential explanatory variables (climate, nutrients,

$CO_2$, disturbances, etc.). After removing the impact of non-climatic factors, the tree-ring index will retain a large amount of climate information [8]. Therefore, dendrochronology has been widely used as a useful technique to study the relationship between tree radial growth and climate change and to further assess impacts of future climate change on forest ecosystems [9].

*Larix* trees are mainly distributed in Northeastern, Northwestern and Southwestern China, and the species have important ecological functions in maintaining forest ecosystems. The three regions experienced rapid warming over the last 40 years in China [10], and the warming could have caused changes in the sensitivity of the species' growth to climate factors, thereby affecting the forests' structure, function and composition [11]. In Northeastern China, temperatures with both positive and negative effects on *Larix olgensis* growth have been detected in Changbai Mountain [12], while summer temperatures controlled *Larix gmelinii* in the Greater Daxing'an Mountains [13]. In Northwestern China, the radial growth of *Larix sibirica* was negatively affected by the January temperature on Altain Mountain [14], and its growth's sensitivity to climate has also changed under the warming [15]. However, the growth response of *Larix* species to climate change has scantly been studied in Southwestern China.

The CHMs are located on the southeastern margin of the Tibetan Plateau and represent an area sensitive to climate change [16]. The CHMs are the source area of many rivers, and their forest ecosystems play an important role in water conservation, soil erosion reduction and biodiversity conservation. In recent years, many studies have been conducted in the area for typical tree species, such as *Picea likiangensis* [17], *Abies georgei* [18], *Pinus densata* and *Pinus yunnanensis*, to detect radial growth responses to climate variables [16]. However, the range of sampling sites was limited. Larger spatial-scale tree-ring studies can better reveal key climatic factors affecting tree growth and explore the ways in which the effects of climate change impact forest ecosystems in the area.

*Larix potaninii* is a main component of subalpine forests and is also one of the timberline tree species in the CHMs, which makes it interesting as a suitable species for a dendrochronology study. The seasonal dynamics of the stem radial growth of *L. potaninii* were detected in the Baima Snow Mountain area, and its radial growth mainly occurred between April and August, with fastest growth in June [19]. The result of a survey of the species population dynamics in the treeline ecotone of the adjacent area did not present significant advances during the past four decades, but the density of seedlings and saplings increased on the northern slope [20]. However, studies on annual growth response to climate variables are rare, hindering our understanding of which climatic factors influence the species' growth in the CHMs.

In this paper, we use tree-ring data from five sites to study the climate–growth relationships of *L. potaninii* at their upper distributional limits in the CHMs. The main aim is to detect key climatic factors that determine radial growth and to explore variations in growth response patterns among the sites. According to the hypothesis that low temperature limits tree growth at the upper distributional limit, we assume that (1) positive effects of temperature on radial growth will be obvious at all sites and (2) variation in response to climate variables will be found due to site-dependence.

## 2. Materials and Methods

### 2.1. Study Area

The CHMs are located on southeastern edge of the Tibetan Plateau, in the transitional zone from the Tibetan Plateau to the Yunnan–Guizhou Plateau (Figure 1). It is the core area of the Hengduan Mountains and is a typical alpine and canyon landform. Three major rivers, the Lancang River, Jinsha River and Nujiang River, originating from the Qinghai–Tibet Plateau, flow through this area. The biological resources are very rich and have a prominent role in the study of global climate change. The vertical distribution of vegetation in this area is obvious, between 2700 and 3800 m a.s.l., mainly dominated by *P. densata* forest and *P. yunnanensis* forest, and the shady slope is spruce forest. At an altitude of

3800–4200 m a.s.l., it is a cold–temperate coniferous forest with *A. georgei*, *L. potaninii* and *P. likiangensis* as the major species. Above 4200 m a.s.l. are alpine shrubs and meadows. *L. potaninii* is a shade-intolerant species and grows well on well-drained and slightly acidic brown forest soil, with a vertical distribution of 2700–4200 m a.s.l.

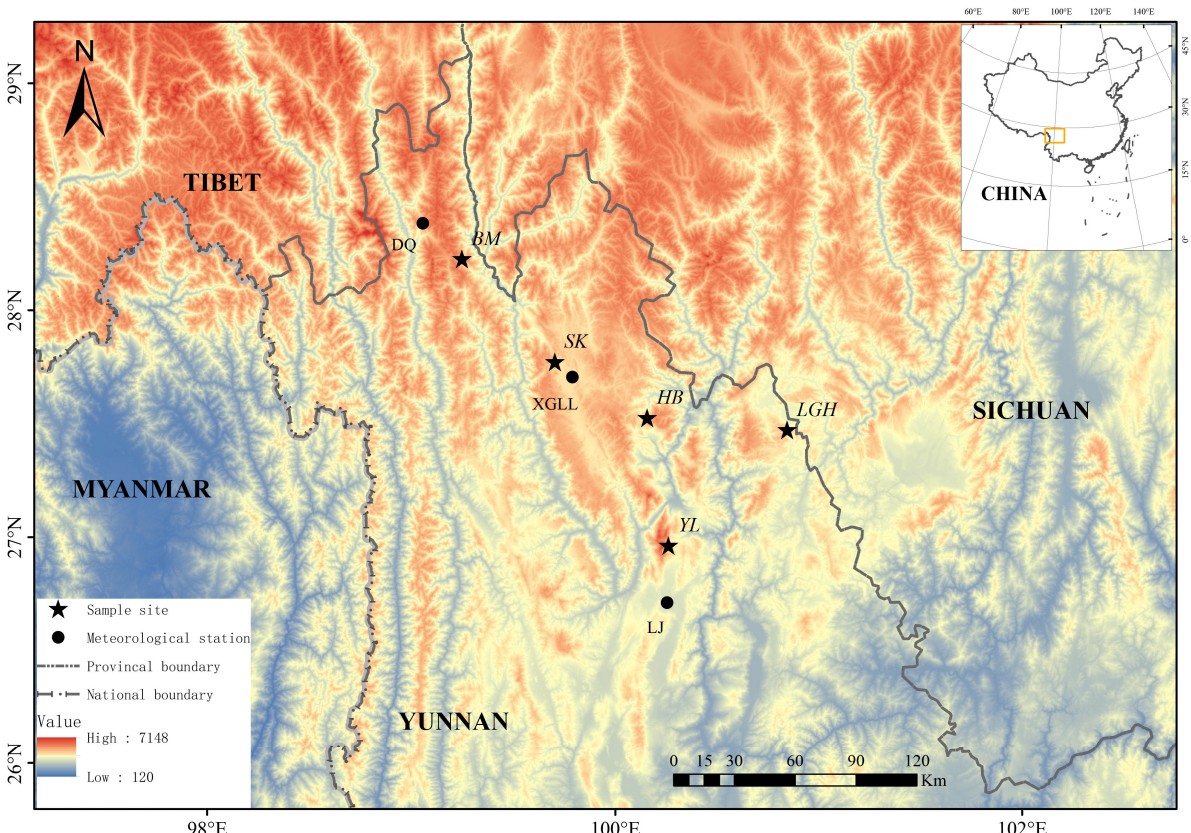

**Figure 1.** Location of the sampling sites and the meteorological stations nearby. BM: Baima Snow Mountain; SK: Shika Snow Mountain; HB: Haba Snow Mountain; LGH: Luguhu Nature Reserve; YL: Yulong Snow Mountain. DQ: Deqin Meteorological Station; XGLL: Shangri-La Meteorological Station; LJ: Lijiang Meteorological Station. The same below.

The CHMs are located in the monsoon climate zone in Southwestern China. The climate has strong seasonal changes due to the interaction of the southwest monsoon, the southeast monsoon and the continental alpine climate of the Qinghai–Tibet Plateau. Summer is short and winter is long, with large daily and annual temperature variations. The climate is divided into wet and dry seasons, and the climate has the characteristics of rain and heat in the same period. According to the instrumental data of Deqin Meteorological Station (1950–2014, Figure 2a), Shangri-La Meteorological Station (1960–2014, Figure 2b) and Lijiang Meteorological Station (1950–2014, Figure 2c), the annual average temperature is 5.6, 5.9 and 12.7 °C, respectively. The precipitation is relatively abundant, and the annual total precipitation is 747, 634 and 965 mm, respectively, mainly concentrated in June–September, accounting for 64%, 73% and 81%, respectively (Figure 2).

Climate data from the global Climatic Research Unit (CRU) show a similar pattern to the three meteorological stations (Figure 2d), and a significant warming trend has been observed in $T_{max}$ (Figure 3a), $T_{mean}$ (Figure 3b) and $T_{min}$ (Figure 3c) over the past six decades. More rapid warming has occurred since 1980, and $T_{max}$, $T_{mean}$ and $T_{min}$ have increased by 0.29 °C, 0.26 °C and 0.23 °C per decade, respectively. Meanwhile, the annual total precipitation has decreased slightly (Figure 3d).

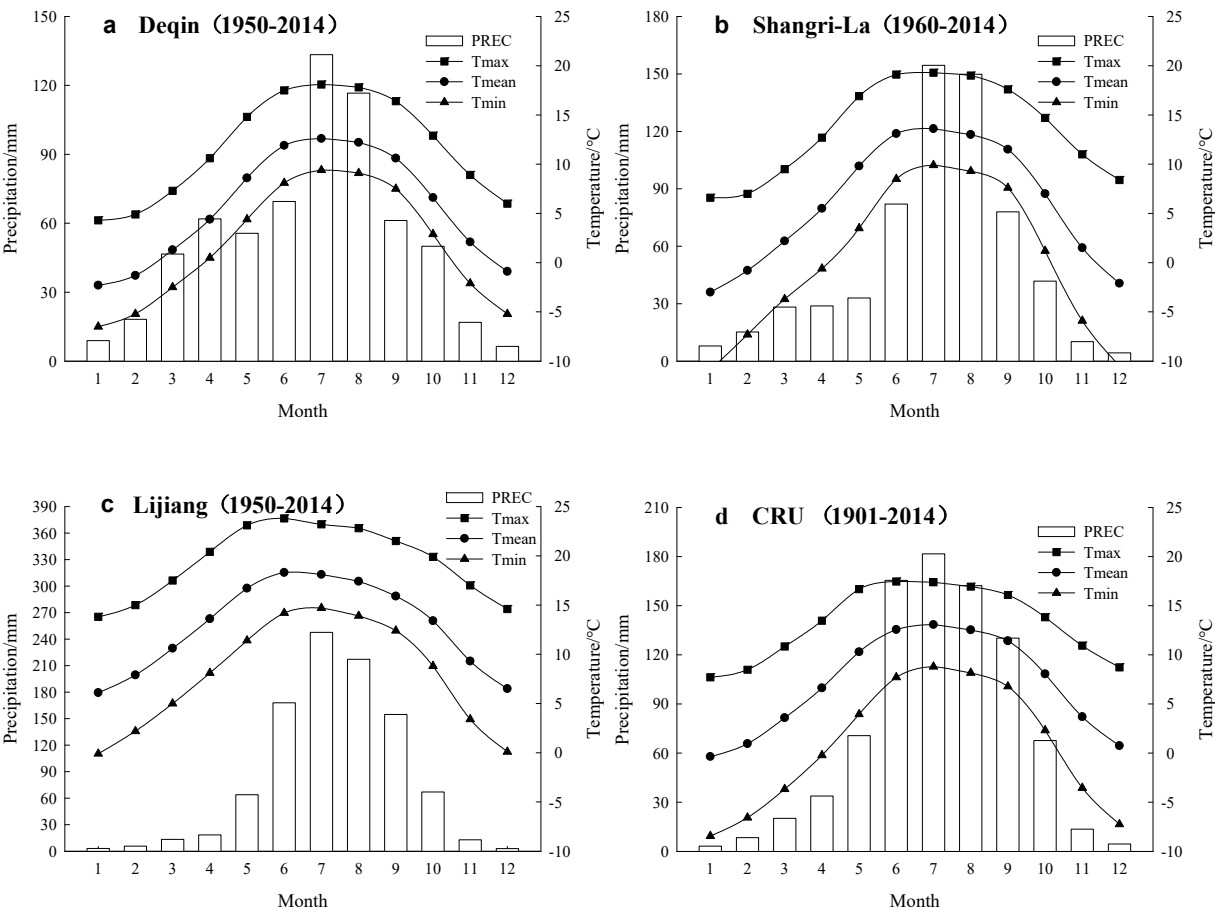

**Figure 2.** Monthly mean temperature and precipitation in Central Hengduan Mountains. $T_{min}$: monthly minimum temperature; $T_{mean}$: monthly mean temperature; $T_{max}$: monthly maximum temperature; P: monthly total precipitation. The same below.

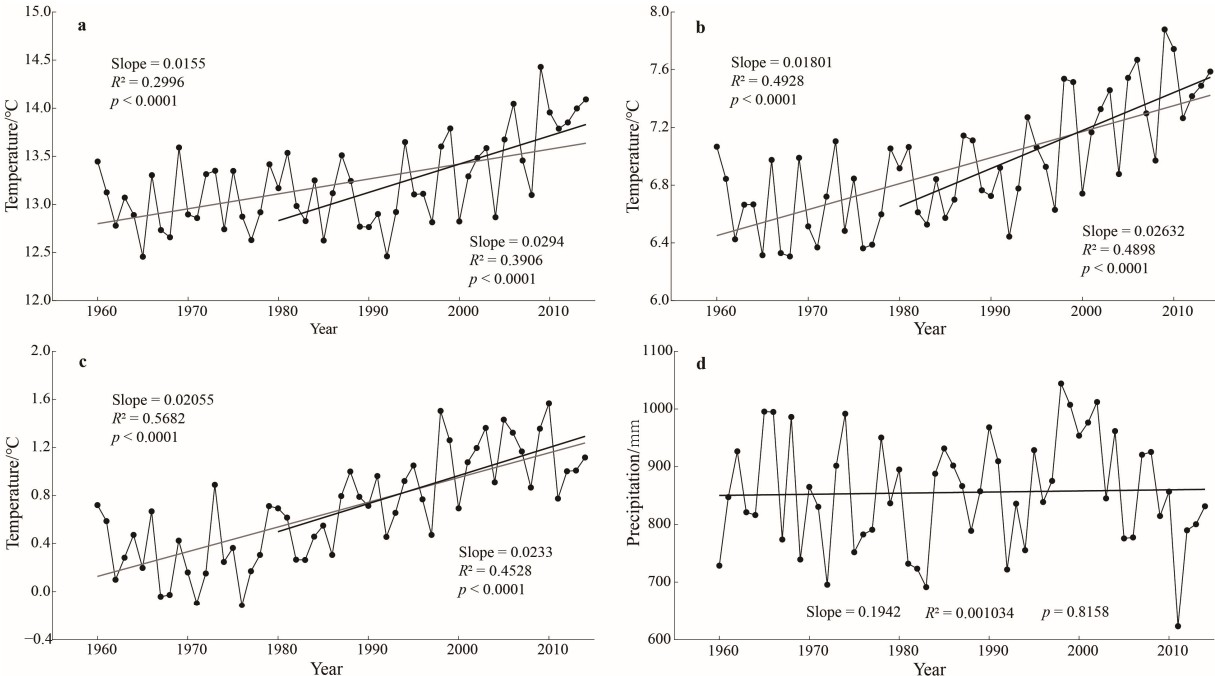

**Figure 3.** Trends of temperature and precipitation in Central Hengduan Mountains during the 1960–2014 period: (**a**): $T_{max}$; (**b**): $T_{mean}$; (**c**): $T_{min}$; (**d**): precipitation.

## 2.2. Field Sampling

In 2015, 2016 and 2021, increment cores of *L. potaninii* were collected at their upper distributional limits of Yulong Snow Mountain, Shika Snow Mountain, Haba Snow Mountain, Baima Snow Mountain and Luguhu Nature Reserve (Table 1). When sampling, older trees with good health were selected, and two cores per tree were taken from opposite directions at a height of 1.3 m. We put the obtained cores into plastic straws and numbered them; a total of 143 trees and 286 cores were collected from 5 sites (Table 1).

**Table 1.** Description of the sampling sites of *Larix potaninii* in CHMs.

| Sites | Altitude/M | Longitude/E | Latitude/N | No.(Tree/Core) |
|-------|-----------|-------------|------------|----------------|
| BM | 4150 | 99°07′18″ | 28°20′02″ | 29/58 |
| SK | 3819 | 99°36′35″ | 27°53′51″ | 23/46 |
| LGH | 3747 | 100°46′56″ | 27°38′10″ | 34/68 |
| HB | 3675 | 100°04′55″ | 27°20′20″ | 31/62 |
| YL | 3647 | 100°12′46″ | 27°06′14″ | 26/52 |

BM: Baima Snow Mountain; SK: Shika Snow Mountain; HB: Haba Snow Mountain; LGH: Luguhu Nature Reserve; YL: Yulong Snow Mountain. The same below.

## 2.3. Tree-Ring Chronology Development

The samples were brought back to the laboratory and pretreated according to the method of Stokes and Smiley [21]. After being air-dried, samples were fixed to wooden mounts and polished, and then, they were visually cross-dated using a binocular microscope. After this preliminary dating, samples were placed on an EPSON Scan (Expression 11000XL) scanner for scanning. The scanning parameters were set to the 24 bit full color image type and 3200 dpi resolution, and the scanned annual ring images were measured with the CDendro and CooRecorder ver. 7.3 software [22]; the accuracy of the system was 0.001 mm. Then, the COFECHA program [23] was used to verify the accuracy of the cross-dating, and the sample cores which were poorly correlated with the main sequence were identified and eliminated. Finally, 143 trees with 265 cores remained. The spline function with a step size of 67% of the sample length was used for detrending using the ARSTAN program [24], to remove the growth trend caused by the tree's own genetic factors and other low-frequency variations induced by competition and disturbances. All detrended series were averaged on a site-by-site basis using the bi-weight robust to produce residual chronologies (Figure 4), which has greater high-frequency information and removes the autocorrelation effect and was thus applied for further analysis.

## 2.4. Climate Data

Climate data (1901–2014) were obtained from global Climatic Research Unit (CRU) grids (http://www.cru.uea.ac.uk/data/, accessed on 2 November 2022) at half-degree spatial resolution (CRUTS 4.06, https://spei.csic.es/database.html, accessed on 2 November 2022), including monthly maximum temperature, monthly mean temperature, monthly minimum temperature and monthly precipitation. In order to test the homogeneity of the climate data, the Mann–Kendall method [25] was used to test the mutation. The test passed the significance level at 0.05, and data were stable and reliable, thus meeting the requirements for the climate–growth relationship analysis.

Correlation coefficients of climate data (from 1960 to 2014) between the CRU and three other meteorological stations (Deqin, Shangri-La and Lijiang) showed significant correlations (Table 2). Therefore, CRU climate data were applied as representative regional data and used for the growth response to climate variables. $T_{max}$, $T_{mean}$, $T_{min}$ and precipitation were downloaded from the CRU grids (http://www.cru.uea.ac.uk/data/, accessed on 2 November 2022) at half-degree spatial resolution.

**Table 2.** Spearman correlation coefficients of climate data between the CRU and three other meteorological stations for the 1960–2010 period. ** $p < 0.01$.

|  | Shangri-La | Deqin | Lijiang |
|---|---|---|---|
| CRU |  |  |  |
| Precipitation | 0.462 ** | 0.495 ** | 0.727 ** |
| $T_{mean}$ | 0.856 ** | 0.830 ** | 0.872 ** |

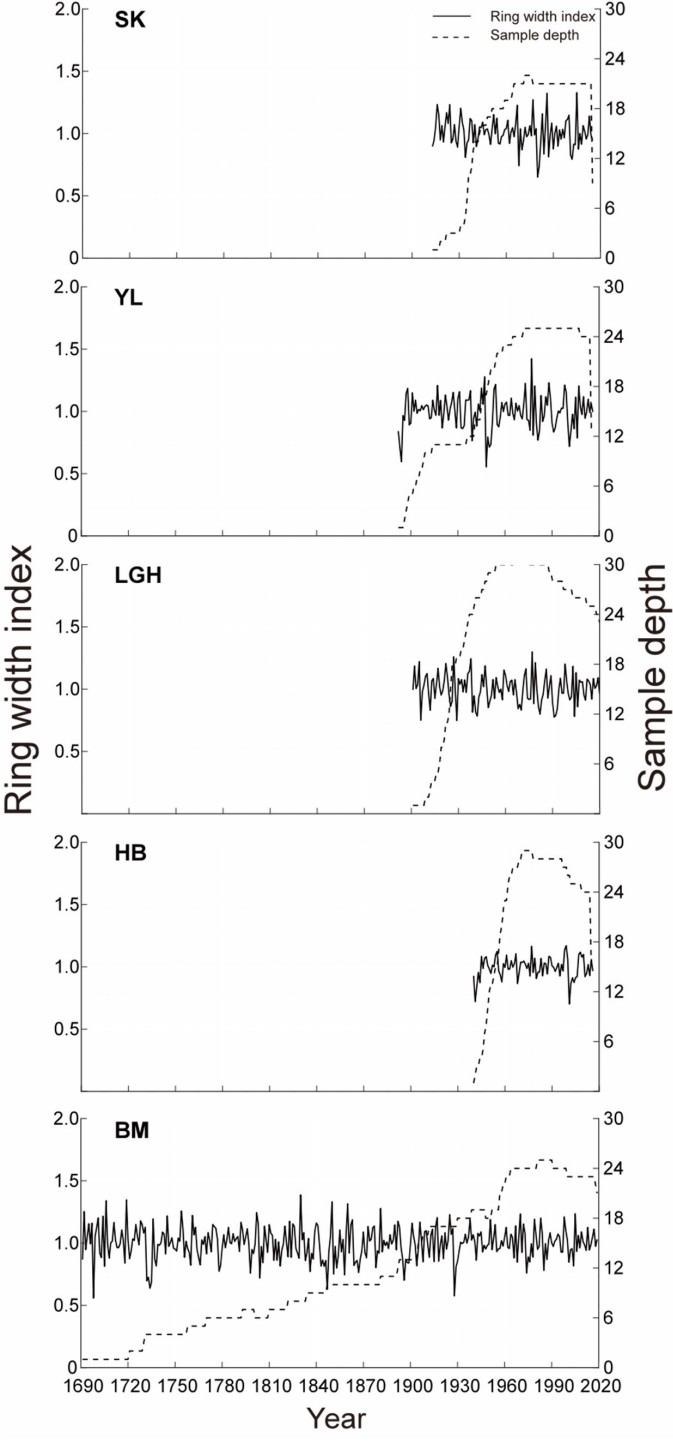

**Figure 4.** Residual tree-ring chronologies and sample depth.

*2.5. Data Analyses*

Statistical analysis of the data was performed using SPSS 19.0 and DendroClim2002 1.0 [26] software. The Spearman method was used in SPSS 19.0 to analyze the correlation among residual chronologies of five sampling points. Considering the influence of the lagged effect of climate on the radial growth, the climatic variables from the previous September to current October were selected. In addition, to evaluate accumulated impacts of climate on tree growth, the seasonalization of dendroclimatic analyses was applied, and the climate variables ($T_{max}$, $T_{mean}$, $T_{min}$ and precipitation) of four seasons were used. According to climate conditions in the CHMs, the four seasons were divided into previous autumn (September–October), current spring (April–May), current summer (June–August) and current autumn (September–October). The response function was applied to detect correlations between residual chronologies and climate variables for the common 1942–2014 period using DendroClim2002. The response function is a linear multiple regression technique that uses the principal components of monthly climatic conditions to estimate the tree-ring growth.

In addition, to better understand growth responses of *L. potaninii* to climate variables, redundancy analysis (RDA) was applied to explore the common signals of climate on tree growth, and this method has been widely used in dendroclimatology studies [27–29]. In the correlation matrix, years were considered as samples, five residual chronologies were considered as response variables and climate variables were considered as explanatory variables. Climate variables were selected using forward selection based on goodness of fit and tested for significance ($p < 0.05$) by using 999 Monte Carlo random permutations, and RDA was conducted using the CANOCO 4.5 program [30].

## 3. Results

The chronologies had high mean sensitivity (MS), variance in first eigenvector (PC1) and correlations between trees (Rbar) and signal-to-noise ratio (SNR) at five sampling sites (Table 3), indicating the high quality of the chronologies and the representativeness of the characteristics of tree-ring width in the area. Of the expressed population signal (EPS), the values of all chronologies exceeded 85%, suggesting that the qualities of the chronologies were good and suitable for the dendrochronology study.

**Table 3.** Statistics of residual chronologies and common interval analyses.

| Residual Chronologies | BM | SK | LGH | HB | YL |
|---|---|---|---|---|---|
| No.(tree/radii) | 29/52 | 23/44 | 34/61 | 31/58 | 26/50 |
| Chronology length | 1687–2019 | 1911–2015 | 1899–2020 | 1937–2016 | 1888–2016 |
| Mean sensitivity | 0.23 | 0.21 | 0.21 | 0.15 | 0.21 |
| Statistics of common interval analysis (1942–2014) | | | | | |
| Variance in first eigenvector/% | 40.87% | 47.03% | 41.98% | 37.29% | 44.52% |
| Standard deviation | 0.21 | 0.18 | 0.19 | 0.13 | 0.19 |
| Signal-to-noise ratio | 16.63 | 22.48 | 26.80 | 27.11 | 29.03 |
| Expressed population signal | 0.94 | 0.96 | 0.96 | 0.96 | 0.97 |
| Rbar | 0.41 | 0.54 | 0.43 | 0.39 | 0.43 |

According to the Spearman correlation coefficients (Table 4) among five chronologies for the common 1943–2014 period (Table 4), all chronologies were significantly ($p < 0.01$) and positively correlated, indicating that the chronologies had high consistency. The degree of correlation of chronology is mainly affected by the distance between the sampling sites. Yulong Snow Mountain is closest to Haba Snow Mountain and has the highest chronological correlation, while Baima Snow Mountain and Luguhu Nature Reserve are the farthest away and have the lowest chronological correlation.

The September $T_{min}$ in the previous year negatively affected the radial growth of *L. potaninii*, with significant correlations at two sites (LGH and YL). The temperature (mainly $T_{mean}$ and $T_{max}$) in current May and June positively affected tree growth by show-

ing significant and positive correlations at four sites (BM, LGH, HB and YL). Meanwhile, precipitation had negative impacts on tree radial growth, showing significant correlations with previous October for SK and LGH; current May for BM, SK, LGH and YL; and current June for BM, HB and YL (Figure 5).

**Table 4.** Correlation coefficients of five residual chronologies among sampling sites for the common 1943–2014 period in Central Hengduan Mountains.

|  | HB | BM | YL | SK |
|---|---|---|---|---|
| BM | 0.467 ** | - | 0.445 ** | 0.436 ** |
| YL | 0.593 ** | 0.445 ** | - | 0.427 ** |
| SK | 0.481 ** | 0.436 ** | 0.427 ** | - |
| LGH | 0.442 ** | 0.331 ** | 0.543 ** | 0.393 ** |

** $p < 0.01$.

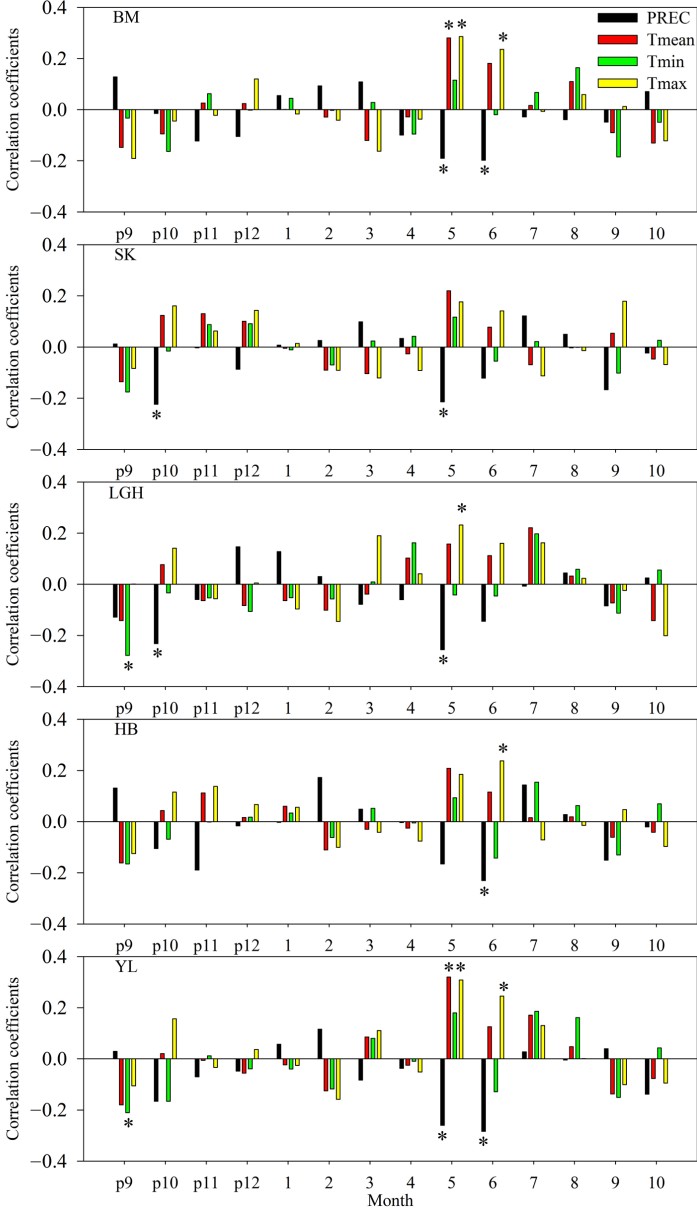

**Figure 5.** Correlation analysis of response function between the residual chronologies and the monthly climatic variables. *: $p < 0.05$. p: previous year. $T_{max}$: monthly mean maximum temperature; $T_{mean}$: monthly mean temperature; $T_{min}$: monthly mean minimum temperature. PREC: precipitation.

For the seasonal dendroclimatic analysis, four sites showed significant correlations between climate variables and tree radial growth (Table 5). Summer temperatures ($T_{max}$ and $T_{mean}$) and current spring $T_{mean}$ positively affected the species radial growth in BM and LGH, respectively, while previous autumn $T_{min}$ and current spring precipitation negatively influenced tree growth in YL and SK, respectively.

For five sites, similar patterns of growth responses to climate were found at the seasonal scale, which also supported the results at the monthly scale (Figure 5). Negative correlations with the precipitation of previous autumn and current spring, negative correlations to $T_{min}$ of previous autumn and positive correlations to $T_{mean}$ of current spring were detected at five sites, although few values showed significance. The positive impact of summer temperatures on tree growth was obvious in BM.

The results of the RDA (Figure 6) were consistent with the results of the response function at monthly and seasonal scales, showing that both temperature and precipitation influenced the radial growth of *L. potaninii* at upper distributional limits in the CHMs. The temperature in current May positively affected tree growth, while precipitation in current May and June, precipitation in previous October and $T_{min}$ in previous September had negative impacts on the radial growth.

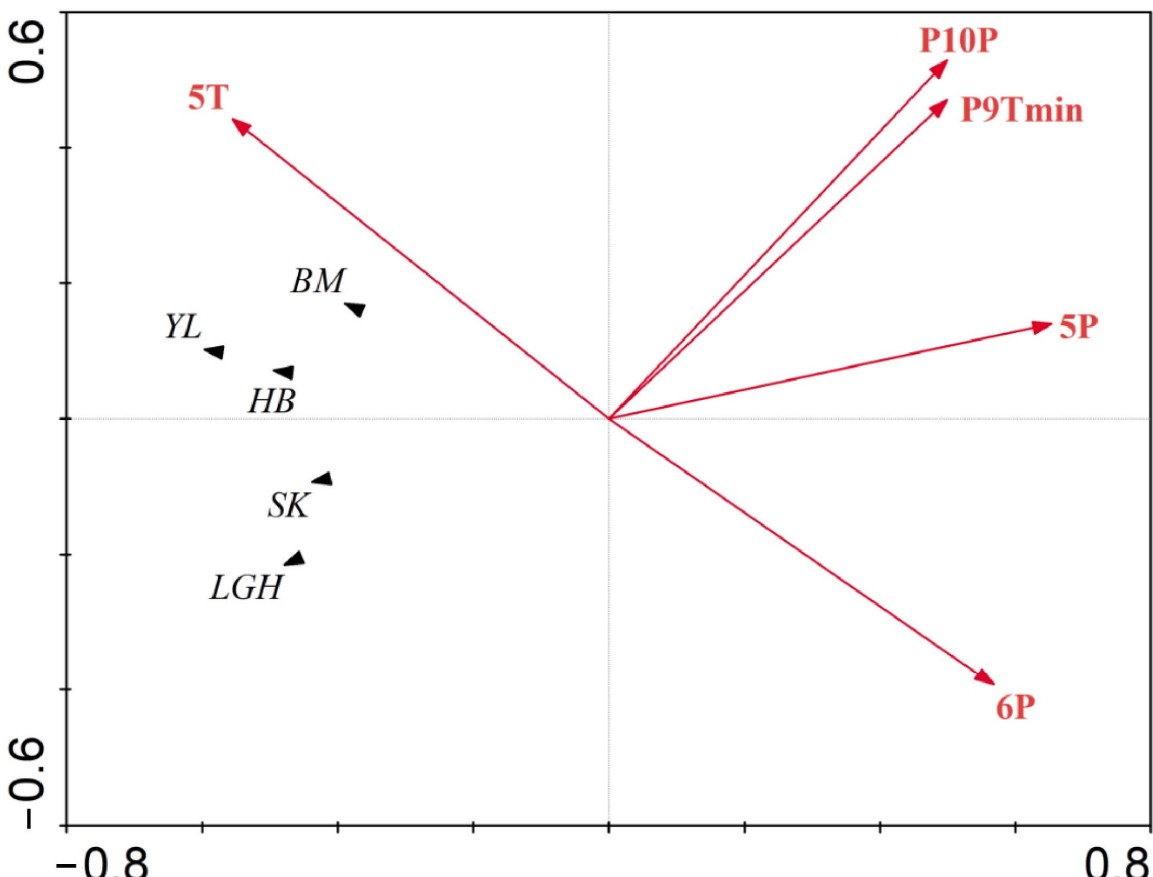

**Figure 6.** The redundancy analysis between five residual chronologies and climatic variables for the common 1942–2014 period. Only significant ($p < 0.05$) climate variables were presented. The longer vector of the climate variable (red lines) indicates the greater contribution. Correlation coefficients between the climatic variables and the chronologies are illustrated by the cosine of the angle between the two vectors. Chronologies and climate vectors pointing in the same direction represent a positive correlation, and the opposite direction indicates a negative correlation. 5T: current May temperature. 5P: current May precipitation. 6P: current June precipitation. P10P: precipitation in previous October. P9T$_{min}$: minimum temperature in previous September.

**Table 5.** Correlation coefficients between residual chronologies and seasonal climate variables.

| Site | $T_{max}$ | | | | $T_{mean}$ | | | | $T_{min}$ | | | | Precipitation | | | |
|------|-----|-----|--------|-------|-------|-------|-------|-------|-------|-------|-------|-------|-------|---------|-------|------|
|      | PAU | CSP | CSU    | CAU   | PAU   | CSP   | CSU   | CAU   | PAU   | CSP   | CSU   | CAU   | PAU   | CSP     | CSU   | CAU  |
| BM   | 0.16 | 0.05 | 0.36 * | −0.06 | −0.06 | 0.02 | 0.23 * | −0.06 | −0.27 | −0.11 | 0.07 | −0.06 | 0.01 | −0.05 | −0.05 | 0.02 |
| SK   | 0.11 | 0.06 | 0.28 | −0.13 | 0.00 | 0.01 | 0.11 | −0.08 | −0.11 | −0.07 | −0.03 | −0.03 | −0.07 | −0.08 * | −0.03 | 0.02 |
| LGH  | 0.09 | 0.17 | 0.01 | −0.22 | −0.11 | 0.16 * | 0.01 | −0.05 | −0.27 | 0.15 | 0.04 | 0.14 | −0.05 | −0.09 | 0.06 | 0.08 |
| HB   | 0.16 | 0.08 | 0.07 | −0.12 | −0.06 | 0.07 | 0.02 | −0.03 | −0.24 | 0.05 | 0.01 | 0.06 | −0.05 | −0.03 | 0.01 | 0.04 |
| YL   | 0.26 | 0.08 | 0.13 | −0.27 | −0.11 | 0.08 | 0.06 | −0.05 | −0.43 * | 0.06 | 0.06 | 0.17 | −0.08 | −0.06 | −0.02 | 0.09 |

PAU: previous autumn; CSP: current spring; CSU: current summer; CAU: current autumn. * $p < 0.05$.

## 4. Discussion

Our results revealed that both temperature and precipitation affected the radial growth of *L. potaninii* at upper distributional limits in the CHMs, which partly supports our hypothesis that low temperature limited tree growth. However, residual chronologies from five sites presented consistent correlations with climate variables, which disproved another hypothesis that variation in growth response would occur between sites. Temperature had both positive and negative impacts on tree growth, while precipitation was negatively correlated with radial growth.

The positive association of *L. potaninii* growth with temperature (either $T_{mean}$ or $T_{max}$) in current May and June was found at four sites, indicating the importance of physiological processes at the early growth stage to the radial growth. In high mountain areas with a cold climate, conifers start xylogenesis at the temperature of 4–5 °C [31]. *L. potaninii* starts wood formation in April at the temperature of 4 °C and reaches its maximum daily growth at the end of June in the study area [19], which indicates June is the major month for radial growth during the year. At the beginning of the growing season, new leaves begin to germinate, xylem cambium cells begin to divide and elongate, and the increase in temperature is particularly important for the radial growth at this time [32]. High temperature was accompanied by sufficient sunlight, which was conducive to photosynthesis and the production of organic matter, therefore promoting radial growth [33]. Positive relationships of *Larix* radial growth with early growing season temperature were also reported in the Hengduan Mountains [34], adjacent high-altitude areas [35] and high elevations of Changbai Mountain [12] and the Great Xingan Mountains [36] in Northeastern China.

The negative impacts of precipitation in current May and June were detected via both the response function and RDA, which helped to explain the indirect effect of temperature on the growth of *L. potaninii*. May and June are months of fast radial growth, and increases in precipitation would also bring an increase in cloudiness, accompanied with a decrease in solar radiation and low temperature, thereby reducing the rate of photosynthesis and the accumulation of organic matter, thus inhibiting the radial growth of *L. potanini* [37]. The negative influence of precipitation on tree growth during the early growing season was also reported for other species in HB [16], SK [17] and BM [18], suggesting the limiting role of May and June precipitation in the study area.

In contrast, some dendrochronology studies reported that spring drought was a limiting factor restricting the radial growth of *A. georgei* and *P. likiangensis* in the Hengduan Mountains [5,37], whereas high precipitation in April–June would enhance tree growth. The reason for this contrast might be related to species-dependent biological characteristics of different species. *L. potaninii* prefers light and heat and grows well in well-drained soil. Melting snow, together with high precipitation in May, could cause waterlogging in the soil which may place limitations on root activity [38] and consequently reduce the photosynthetic rate and limit tree growth. However, *P. likiangensis* and *A. georgei* have characteristics of higher water demand and tolerance to waterlogged conditions as compared to *L. potanini*, so high precipitation would reduce drought stress; therefore, these species show positive responses to spring precipitation.

$T_{min}$ in the previous September was found to negatively affect tree growth. This may reflect physiological responses at night-time. The minimum temperature usually occurs at night and high temperature could accelerate the respiration rate, leading to the excessive consumption of nutrients stored in stems and resulting in a negative impact on radial growth for the coming year [39]. A study by Zhang et al. (2022) reported that the early cambium activity of *L. potanini* uses the non-structural carbohydrates accumulated in the previous year [19]. $T_{min}$ was observed to be a dominant factor affecting tree growth in high-elevation fir forests in the Southeastern Tibetan Plateau [40], and negative relationships between previous post-growing season temperatures and *P. likiangensis* growth were reported in SK [17].

October precipitation in the previous year negatively affected radial growth. October represents the end of *L. potanini*'s growth stage, and cambial growth was slow [19]. Su-

perabundant precipitation would reduce the oxidation reduction potential in the soil and decrease the accumulation of nutrients by limiting the activity of the roots [38]. Moreover, more precipitation means lower temperature, which may reduce photosynthetic activity, consequently preventing the formation of wide ring widths in the next year.

The positive influence of early summer (June) temperature on radial growth was detected at five sites, with three (BM, HB and YL) presenting significant correlations. But for the whole summer, tree growth only significantly responded to temperatures in BM, specifically to $T_{max}$ and $T_{mean}$. The reason for this may be related to site location. BM is in the highest and northernmost point among the five sites, so its environment is harsher, and growth is more sensitive to climate change. Therefore, an increase in temperature in the summer (growing season) may be particularly important for the formation of wide rings. Particularly, $T_{max}$ normally represents the temperature of day-time, and a higher $T_{max}$ would promote photosynthesis and thus enhance radial growth. Similar results were also reported for *A. georgei* in the same area [18], in the adjacent Southeastern Tibetan Plateau [40] and Southern Tibetan Plateau [41] and for *L gmelinii* at high altitudes in Northeastern China [36].

In our study area, annual mean temperature is predicted to increase significantly while annual precipitation will slightly decrease according to EC-EARTH based on the CMIP 5 under the RCP 4.5 scenario [42]. Particularly, the most obvious increases are predicted in April and May for temperature and in July and October for precipitation. If positive influences induced by increases in May and June temperature could offset the negative effects of increases in temperature of September and precipitation of May, June and October, future warming would enhance *L. potanini*'s radial growth at the upper distributional limit in the CHMs and may cause an upward shift in the species distribution range. Although warming is generally accompanied with increases in evaporation in soil, causing drought stress and ultimately inhibiting tree growth and seed germination [43], our study did not show a water deficiency of *L. potanini* growth, and the sites were not prone to drought conditions. A previous study found that the treeline positions of the Hengduan Mountains have not significantly changed since 2000, but plant density has significantly increased at the treeline margin [44], and an upward shifting trend of the *L. potaninii* treeline in China was reported by Mamet et al. (2019) [45].

## 5. Conclusions

The radial growth of *L. potaninii* at the upper distributional limit in the CHMs showed a consistent response to climate. Warm and thermal conditions in the early growing season (May and June) are important to the species' radial growth, and high temperatures and sufficient heat are also beneficial. Higher temperatures ($T_{min}$ of September) and more precipitation (October) in the previous post-growing season inhibit radial growth, showing the lag effect of climate on tree growth. According to the results of growth sensitivity to climates, we expect that the radial growth of *L. potaninii* would benefit from future warming if the increasing temperature of May could offset the negative influences of excessive precipitation.

**Author Contributions:** H.Y. and Y.Z. conceived the study and wrote the paper. H.C. and S.X. designed the study and performed the experiment. K.T. and D.Z. analyzed the data. J.L. and M.S. contributed materials and analysis tools. All authors have read and agreed to the published version of the manuscript.

**Funding:** This research was supported by the National Natural Science Foundation of China (No. 31600395) and the Key Program of the Educational Department of Yunnan Province (No. 2015Z136).

**Data Availability Statement:** The data presented in this study are available on request from the corresponding author.

**Acknowledgments:** We would like to acknowledge the administrative organizations of the five sampling sites for providing permission to collect tree-ring samples.

**Conflicts of Interest:** The authors declare no conflict of interest.

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
