# Peer review of "Relationships between Climate Variability and Radial Growth of Larix potaninii at the Upper Altitudinal Limit in Central Hengduan Mountain, Southwestern China"

_forests, doi:10.3390/f14091790_

Round 1
Reviewer 1 Report
Interesting dendroclimate research on a deciduous conifer in China. Some comments:
Figure 4: I was surprised by how different the top chronology appeared versus the others. Then I noticed that the x-axis of that top plot is much longer than the others. Somehow, I would match up the x-axes to be closer in time range. Site BM is clearly different from the others by being 250 years older. Also, dendrochronologies are easier to view and study when a horizontal reference line, 1.0 in this case, is overlay plotted with them. And, sample depth appears to be in terms of # of cores, but in dendrochronology the basic unit of sampling is the tree, not the number of cores within given trees. Clearly, # of trees and # of cores are related (perfectly, apparently, from Table 1), but still it would be more honest to convey sample depth as # of trees.
Table 3: Given that these stats are of the processed chronologies, it would seem that mean sensitivity and standard deviation are redundant to each other. Plus, SNR is notoriously hard to interpret because it is influenced by the number of samples included. A better indicator of chronology strength would be the classic Rbar value, commonly given in the ARSTAN output. Rbar is not influenced by sample size, so it is a better indicator of chronology strength.
Figure 5: Only monthly correlations? No seasonalizations? Doesn’t DendroClim2002 provide seasons composed of multiple consecutive months? Seasonalization of dendroclimate analysis can reveal climate influence not necessarily obvious at the monthly scale.
Conclusions: I suppose it’s possible that some trees respond favorably to reduce moisture availability and/or might benefit from future warming, but these interpretations would be counterintuitive from a pure forestry perspective. At a minimum, both interpretations probably have a limit to them: if it truly got hot and dry at these sites, surely even these trees would begin to struggle?
good, only minor edits needed
Author Response
Dear reviewer
Thanks for your consideration of our submission (forests-2447332) and valuable comments in improving the MS. After reading the comments, we found those suggestions very helpful and corrected accordingly. Please find our revised paper together with detailed responses to referees' comments. Original comments made by the referee are in italics and our responses are in regular font. In the text of the revised paper, we mark all changes in track version.
We hope that the new MS would meet the high publication standards of Forests.
Sincerely
Haitao Yue & co-authors
Kunming
2023-08-29

Reviewer 2 Report
The manuscript submitted for review, entitled 'Relationships between climate variability and radial growth of Larix potaninii at the upper altitudinal limit in Central Hengduan Mountain, Southwestern China', presents an analysis of the response of tree growth to climatic conditions for five sites. The introduction briefly and accessibly describes the local situation and conditions of the study. Standard analyses were applied and older statistical packages were used, however, sufficient to answer the stated purpose of the work. The results are described in a readable manner, and the discussion briefly relates to the results. Because of the research area, the results on Larix potaninii bring some novelty and are worth publishing.
The manuscript is well prepared. I only have minor comments and suggestions:
In lines 146-147 and 161-163 the same content is repeated. Please correct this.
In the description of Figure 2, it is worth adding the period to which the data refers even though the dates are in the text.
Figure 6 - only a portion of the figure is visible (the right and bottom edges are not visible). Please improve the visibility of the figure and put the complete image.
324 The Conclusion makes the statement ...The species did not show a drought stress... which is too risky. No drought indices (e.g., SPEI) were included in the paper. Negative correlations with precipitation are not a proof of lack of drought stress. Please reconsider this sentence.
Suplement
Please consider adding a supplement with raw chronologies. They provide accurate information about Larix potaninii growths. Perhaps these data are familiar to local readers, while for people from other parts of the world this information is unknown, and raises curiosity. The RWI value itself is important from a statistical point of view, while the actual increment widths are important from an illustrative point of view. They can be placed on a graph in a supplement or in the form of a statistical table. A manuscript without these data is valuable, but much poorer and less pleasant to read. Raw values can also affect the higher citation rate of a paper.
Author Response

(The authors gave the same response as above.)

Reviewer 3 Report
After careful evaluation, I have decided that your manuscript requires major revisions before it can be considered for publication.
Strengths:
- The study addresses an important ecological question, investigating the relationships between climate variability and radial growth of Larix potaninii in the context of future climate change.
- The use of dendrochronological methods to assess the influence of climatic factors on tree radial growth is valuable and provides a solid scientific approach.
- The study contributes to the understanding of forest ecosystem responses to climate change, particularly in subalpine regions.
Major Revisions Required:
- Clarity and Structure: The article's structure needs improvement to enhance clarity and logical flow. Consider reorganizing the sections to present a more coherent narrative that guides the reader through the research process, methods, results, and implications more effectively.
- Abstract: The abstract should provide a concise yet comprehensive overview of the study's objectives, methods, key findings, and implications. Ensure that it accurately reflects the content of the article.
- Methodology: Provide a more detailed description of the dendrochronological methods used, including information on sample collection, processing, and statistical analyses. This will help readers understand the robustness of your results.
- Results and Discussion: Clarify the presentation of results and link them more explicitly to the research questions. Discuss the implications of each result and how they contribute to the broader understanding of climate-forest interactions. Additionally, consider integrating more contextual information or comparisons with relevant existing studies.
- Interpretation of Results: Provide a more thorough interpretation of the findings in the context of existing literature. Explain why certain climatic factors have positive or negative impacts on radial growth, considering potential ecological mechanisms or interactions.
- Language and Style: The manuscript requires substantial language editing for clarity, conciseness, and proper grammar. Ensure that scientific terminology is used accurately and consistently throughout the article.
- Figures and Tables: Review and refine the presentation of figures and tables to enhance their clarity and readability. Ensure that each figure/table is appropriately labeled and explained in the main text.
- Conclusions and Implications: Provide a well-structured conclusion that summarizes the key findings and discusses their broader ecological and practical implications. Highlight the significance of your results for predicting forest responses to future climate change.
- Keyword Choice: Consider revising and expanding the list of keywords to include terms that better represent the study's scope and content.
We appreciate the valuable contribution your study makes to the field of dendrochronology and its implications for understanding climate-forest interactions. Addressing the above-mentioned points in your revision will significantly enhance the clarity, robustness, and overall impact of your manuscript.
The manuscript requires substantial language editing for clarity, conciseness, and proper grammar. Ensure that scientific terminology is used accurately and consistently throughout the article.
Author Response

(The authors gave the same response as above.)

Round 2
Reviewer 3 Report
Respected Authors,
I would like to commend your diligent work in revising the manuscript to align with the suggested comments. Your responsiveness and commitment to improving the paper's quality are commendable. Following a comprehensive evaluation of the revised manuscript, I have observed significant advancements that render it suitable for publication.